# Quality of life and retention in care among people living with HIV initiated on ART in the era of "Universal Test and Treat" policy at a large HIV Clinic in South Western Uganda

**Faith Tumuhairwe[1], Jonathan Izudi[2]\*, Abel Munina[1], Anthanasio Bashaija[3], Francis Bajunirwe[2]**

1  The AIDS Support Organization (TASO), Mbarara, Uganda, 2  Department of Community Health, Faculty of Medicine, Mbarara University of Science and Technology, Mbarara, Uganda, 3  Department of Foundations of Education and Psychology, Kabale University, Kabale, Uganda

\* jonahzd@gmail.com

## Abstract

### Background

Anti-retroviral therapy (ART) improves the quality of life (QoL) among people living with human immunodeficiency virus (PLWH). Most studies documenting the gains in QoL have been conducted among persons starting treatment at advanced HIV disease. In the era of Universal Test and Treat (UTT) policy, most PLWH start ART early with high CD4 counts. Therefore, we investigated the association between baseline CD4 count with QoL and retention among PLWH during UTT in southwestern Uganda.

### Methods

Between June 11, 2019, and June 10, 2020, we reviewed medical records for PLWH initiated on ART between April 2017 and September 2018 and interviewed them to collect QoL data. The primary exposure was CD4 count at ART initiation categorized as <500 cells/µl (low) versus ≥500 cells/µl (high). Physical and mental health-related QoL were the primary outcomes. Retention was the secondary outcome. Binary logistic regression was used to assess the association between the exposure and the primary outcome, while the Cox Proportional Hazard regression model was used for the secondary outcome.

### Results

Of 300 participants, 59.7% initiated ART at a low baseline CD4 count. ART initiation at a higher baseline CD4 count, compared to a lower baseline CD4 count, was associated with a lower mental health-related QoL (adjusted odds ratio [aOR] 0.56,

**Data availability statement:** All relevant data are within the paper and its Supporting information files.

**Funding:** The author(s) received no specific funding for this work.

**Competing interests:** The authors have declared that no competing interests exist.

95% confidence interval [CI] 0.32–0.97) but similar physical health-related QoL (aOR 0.54, 95% CI 0.10–2.78) and retention (adjusted hazard ratio [aHR] 0.30, 95% CI 0.08–1.14).

## Conclusion

In this cohort of PLWH on ART, those who initiated the treatment at a higher baseline CD4 were less likely to have good mental health-related QoL functioning compared to those initiating at lower CD4 counts. However, the two groups were comparable in physical health-related QoL and retention in care. PLWH initiating ART at a higher baseline CD4 may require mental health-related support as part of treatment.

---

## Introduction

Globally, 39 million people were living with human immunodeficiency virus (PLWH) by the end of 2022 [1]. Human Immunodeficiency Virus (HIV) is a preventable and treatable condition but without a cure so requires life-long antiretroviral therapy (ART). The widespread scale-up of ART has transformed HIV into a manageable chronic health condition, allowing PLWH to achieve life expectancy and health outcomes similar to those in the general population. The goal of ART is to improve both the longevity and quality of life (QoL) of PLWH, including preventing HIV transmission [2]. Untreated HIV leads to a high viral load and progression to developing several opportunistic infections, including Acquired Immunogenicity Syndrome (AIDS) [3].

HIV treatment guidelines have evolved, progressively expanding ART eligibility for PLWH and ultimately leading to universal access. Earlier guidelines recommended ART initiation based on the presence of AIDS-defining illnesses or a baseline CD4 count of ≤200 cells/μL, which was later revised to ≤350 cells/μL [4]. In 2013, the World Health Organization (WHO) revised its HIV treatment guidelines to recommend initiating antiretroviral therapy (ART) at a baseline CD4 ≤ 500 cells/mm$^3$. In 2017, Uganda adopted the Universal Test and Treat (UTT) policy into its national HIV treatment guideline, allowing PLWH to start ART regardless of baseline CD4 counts or WHO clinical stage [5,6]. UTT policy is a fast-track strategy aiming at contributing to the United Nations Joint AIDS Programs (UNAIDS) target of ending the HIV epidemic by 2030. Early ART initiation, regardless of immune and clinical status, has multiple benefits. Research shows that PLWH with an undetectable viral load cannot transmit HIV, a concept known as 'Undetectable = Untransmittable' [3,7,8]. Potential concerns around the UTT policy include its impact on quality of life (QoL) and retention, as long-term exposure to ART may lead to adverse effects such as drug toxicities and probable drug interactions among PLWH [9–11]. The side effects may be tolerated poorly, more especially by PLWH who initiate ART at higher CD4 counts possibly leading to decline in QoL and contributing to loss-to-follow up. Few studies have examined QoL and retention in care among PLWH who start ART early—when baseline CD4 count is high—under the UTT policy. Most studies have examined QoL when ART is started late, when the CD4 count is low, and when patients are

symptomatic, often resulting in a marked decline on the QoL domains. Therefore, it is not completely clear whether ART initiation at a higher baseline CD4 count might lead to negative effects on QoL or retention compared to ART initiation at a lower baseline CD4 count among PLWH in resource-limited settings such as Uganda.

Accordingly, we compared the physical and mental health-related QoL and retention between adult PLWH who initiated ART at a higher versus lower baseline CD4 count in rural Uganda. We primarily hypothesized that in the era of UTT policy, QoL scores for PLWH initiated on ART at a higher baseline CD4 count might be lower hence lower retention in care compared to those who initiated ART at a lower baseline CD4 count. The secondary hypothesis was that retention would be lower among PLWH with higher baseline CD4 counts compared to those with a lower baseline CD4 count at ART initiation, as they perceive themselves healthier or less "sick". Consequently, they may be more likely to discontinue ART due to adverse effects following initiation. Evidence from this study will inform clinical care and interventions required to optimize QoL outcomes among PLWH initiating treatment before becoming symptomatic.

## Methods and materials

### Study design and setting

First, we utilized a retrospective design to review medical records for PLWH who had initiated ART under the UTT policy between April 2017 and September 2018, to obtain data on CD4 count when treatment started. Second, we used a cross-sectional study design to collect the data on the current status of QoL from the PLWH whose records had been reviewed in the retrospective study. Data were collected between June 11, 2019, and June 10, 2020. The study was conducted among PLWH receiving HIV care at The AIDS Support Organization (TASO) HIV Clinic in Mbarara district in Southwestern Uganda. TASO is a non-governmental and one of the oldest and the first HIV organizations to respond to the HIV pandemic in Uganda and sub-Saharan Africa. TASO has 11 service centers spread across Uganda and one training center (TASO College of Health Sciences) located in Kampala, the capital city of Uganda [12,13]. TASO Mbarara was established in 1989 and became a semi-autonomous Centre in 1991. Located in Mbarara City adjacent to the Mbarara Regional Referral Hospital, TASO Mbarara has enrolled over 6,000 PLWH, both adults and children in care by June 2023. HIV treatment using ART started in 2005 and the HIV Clinic adheres to the Uganda National HIV Treatment Guidelines.

### Study population

The study participants consisted of PLWH aged 18 years and above receiving ART under UTT policy between April 2017 and September 2018. The participants were identified retrospectively through the clinic records and then consecutively sampled until the desired sample size was reached.

### Variables and measurement

The primary outcomes were physical and mental health-related QoL at the time of the interviews and retrospectively 3 months before the interview. We measured QoL using the Medical Outcome Study-HIV Health Survey (MOS-HIV) tool. The MOS-HIV is the most commonly used questionnaire for the evaluation of health-related QoL in PLWH in both medical settings and research studies [14]. The scores range from 0 to 100. MOS-HIV is a culturally adapted tool and has been shown to have good psychometric properties in several settings [15,16]. The average coefficient of 0.91 for the total score and coefficients above 0.80 for most subscales, except role functioning, which has an average score of 0.76 makes the MOS-HIV good for PLWH. It is a 35-item questionnaire, with 10 dimensions that focus on health perceptions, pain, physical, role, social and cognitive functioning, mental health, energy, health distress, and QoL, and it takes approximately 5 minutes to complete. Its established validity and reliability make it a robust tool for assessing QoL in this population.

The participants rated their QoL using the five Likert scale and the responses ranged from excellent to poor denoted by values that ranged from five to one for each question. We summarized the QoL scores and standardized them to achieve

a mean of 50±6. We then computed two summary scores, the physical health summary and the mental health summary [17], with higher scores indicating better functioning. We categorized the physical and mental health-related QoL based on the median score as low when the scores were less or equal to the median, and high when the scores were above the median. While no universally accepted threshold exists for a clinically meaningful difference in MOS-HIV scores, prior research on similar Health-Related Quality of Life (HRQoL) tools suggests that differences in the range of 5–10 points may be considered clinically significant.

The secondary outcome was retention measured on a binary scale (yes vs. no) for those active in care or lost to follow-up. Participants who were active in care up to 7 days or less (≤7 days) after their last clinic appointment date for ART refills were considered retained whereas those who had missed their last clinic appointment date for ART refills by more than 7 days (>7 days) were considered lost to follow-up hence not retained in care. We applied censoring to those who were reported as dead during the follow-up time. The primary exposure was the baseline CD4 count, categorized as low when the CD4 count was <500 cells/µl versus high when ≥500 cells/µl. The covariates included age, sex, marital status, occupation, WHO clinical staging at ART initiation, and recruitment site.

## Data collection

The tool was administered to the participants by trained research assistants and interviews were done in a conducive and quiet place within the health facility. The data were collected in both the local *Runyankore* and English languages based on the preference of the respondent. The participants rated their health status at the time of the data collection and retrospectively at 3 months ago using a *'then test'* question, an approach that avoids response shift bias associated with longitudinal assessment of self-reported health outcomes. Interviews with participants took place between June 11, 2019, and June 10, 2020.

## Statistical methods

**Sample size estimation.** We hypothesized that the mean QoL scores for PLWH initiated on ART at a higher CD4 count would be lower than for those initiated at a lower CD4 count. Based on a previous study comprising a mixed cohort of PLWH initiating ART at various CD4 counts, the mean QoL score was 50. We assumed mean scores of 51±6.15 for PLWH initiated on ART at low baseline CD4 count and 49±6.15 for those initiated at high CD4 count. Using a two-sided Student's t-test at a 5% significance level, we computed that about 150 participants would be needed per group (overall n=300 participants) for the study to have 80% statistical power in detecting a real association in the population [18].

**Data analysis.** We used descriptive statistics to summarize numerical data such as age and baseline CD4 count, using the mean and standard deviation (SD) when normally distributed and the median and interquartile range when skewed (S1 File). We summarized categorical data as frequencies and percentages. We assessed differences between the primary outcome and categorical variables using the Chi-squared test for cell frequencies ≥5 or Fisher's exact test for cell frequencies <5. For numerical data, we assessed mean differences in the outcome using the Student's t-test for normally distributed numerical data and the Wilcoxon-rank sum test for skewed numerical data.

We employed a binary logistic regression model to determine the association between baseline CD4 count measured on a binary scale and QoL with and without adjusting for statistically significant and clinically relevant variables, reported as odds ratio (OR) and 95% confidence interval (CI). To evaluate the goodness-of-fit of the logistic regression model, we applied the Hosmer-Lemeshow test. The null hypothesis was that the model provides an adequate fit to the data, while the alternative hypothesis was that the model does not adequately fit the data.

To compare differences in retention rates by the baseline CD4 category at ART initiation, we plotted Kaplan Meier curves and tested observed differences in retention using the log-rank test at a 5% level of significance. To determine the association between baseline CD4 count and retention, we conducted survival analysis using the Cox Proportional Hazard regression model with and without adjustment for statistically significant and clinically relevant variables. We reported

hazard ratios (HR) and its 95% CI. We used Schoenfeld residual testing to check the assumptions of the Cox Proportional Hazard regression model for each variable. The null hypothesis was the model is correctly specified so p > 0.05 was taken to mean no violation of the assumptions.

### Ethical consideration

We obtained ethical approvals from the Mbarara University of Science and Technology Research Ethics Committee (MUST-REC reference number MUREC 1/7). We obtained administrative clearance from the institutional review committee at TASO Headquarters and TASO Mbarara. We explained the study's scientific rationale, social value, approach, informed consent process, confidentiality, privacy, risks, withdrawal, and compensation among others.

This was followed by all participants providing voluntary written informed consent without coercion or undue influence. We maintained privacy by collecting data in a private and secure room within TASO Mbarara and used unique identifiers. Overall, we followed all the ethical principles as stipulated in the Declaration of Helsinki.

## Results

### Socio-demographic and clinical characteristics of participants

We enrolled 300 participants and Table 1 shows their characteristics displayed by the baseline CD4 count. Of the 300 participants, 59.7% were initiated on ART at baseline CD4 count <500 cells/µl while 40.3% were initiated on ART at baseline CD4 count ≥500 cells/µl. Of them, 57.7% were female, 57.3% were aged 30–49 years, and 55.67% were married. The overall mean age was 34.1 ± 9.4 (years), with no statistically significant difference between those initiated on ART with baseline CD4 count <500 cells/µl and those with baseline CD4 count ≥500 cells/µl (33.9 ± 9.1 years vs. 34.4 ± 9.9 years, p = 0.470). We found no statistically significant differences in sociodemographic and clinical variables between participants who initiated ART at baseline CD4 count <500 cells/µl and those who initiated ART at baseline CD4 count ≥500 cells/µl.

### Comparison of QoL with baseline CD4 count

Of the 300 participants, 92.3% responded to the physical health-related QoL domain while 81 56.1% responded to the mental health-related QoL. Of the participants with CD4 count ≥500 cells/µl (Table 2), 86 (76.1%) had good physical health-related QoL while 25 (78.1%) had good mental health-related QoL. Regarding participants with CD4 < 500 cells/ µl, 105 (64.0%) had good physical health-related QoL while 47 (95.9%) had good mental health-related QoL. Retention was higher among participants who were started on ART at baseline CD4 < 500 cells/ul compared to those started on ART at baseline CD4 count ≥500 cells/ul (95.0% vs. 84.3%, p = 0.002). The median retention time was 17 months (IQR 8–19) for individuals with baseline CD4 < 500 cells/cm³ and 18 months (IQR 9–20) for those with baseline CD4 ≥ 500 cells/cm³.

### Association between baseline CD4 count with QoL and retention in care

After adjusting for statistically significant and clinically relevant variables (Table 3), our data showed no statistically significant difference in physical health-related QoL between participants who had ART initiation at a higher baseline CD4 count compared with those who had ART initiation at a lower baseline CD4 count. However, participants with a higher baseline CD4 count were less likely to have good mental health-related QoL compared with those with a lower baseline CD4 count. The Hosmer-Lemeshow test for goodness-of-fit demonstrated that the model adequately fit the data for the physical QoL ($\chi^2$ = 29.14, p = 0.085) and the mental QoL ($\chi^2$ = 113.02, p = 0.8557), indicating no significant discrepancy between the observed and predicted outcomes.

Retention in HIV care was less likely for participants with a higher baseline CD4 count compared to those with a lower baseline CD4 count. Schoenfeld residual testing confirmed that the assumptions of the Cox Proportional Hazards regression model were upheld, as all p-values were greater than 0.05.

**Table 1. Socio-demographic and clinical characteristics of participants.**

| Variables | Level | Overall (n = 300) | CD4 < 500 cells/ µl (n = 179) | CD4 ≥ 500 cells/ µl (n = 121) | P-value |
|---|---|---|---|---|---|
| Sex | Female | 173 (57.7) | 104 (58.1) | 69 (57.0) | 0.853 |
| | Male | 127(42.3) | 75(41.9) | 52(43.0) | |
| Age (years) | <30 | 108 (36.0) | 64 (35.8) | 44(36.4) | 0.890 |
| | 30-49 | 172 (57.3) | 104 (58.1) | 68(56.2) | |
| | ≥50 | 20 (6.7) | 11 (6.1) | 9 (7.4) | |
| | Mean (SD) | 34.1 (9.4) | 33.9 (9.1) | 34.4 (9.9) | 0.470 |
| Marital status | Never married | 54 (18.0) | 30 (16.8) | 24 (19.8) | 0.060 |
| | Married | 167(55.7) | 95 (53.1) | 72 (59.5) | |
| | Divorced/ separated | 69 (23.0) | 50 (27.9) | 19 (15.7) | |
| | Widowed | 10 (3.3) | 4 (2.2) | 6 (5.0) | |
| Religion | Anglican | 150 (50.0) | 87(48.60) | 63(52.1) | 0.599 |
| | Catholic | 100 (33.3) | 59 (32.96) | 41(33.9) | |
| | Others | 50 (16.7) | 33 (18.44) | 17(14.0) | |
| Occupation | Paid employee | 34 (11.3) | 20 (16.5) | 14 (11.6) | 0.745 |
| | Casual laborers/vendor | 143 (47.7) | 87 (48.6) | 56 (46.3) | |
| | Defendant | 55 (18.3) | 35 (19.5) | 20 (16.5) | |
| | Peasant | 68 (22.7) | 37 (20.7) | 31 (25.6) | |
| Recruitment site | Community | 258 (86.0) | 152 (84.9) | 106(87.6) | 0.511 |
| | Facility | 42 (14.0) | 27 (15.1) | 15(12.4) | |
| WHO clinical stage (n = 275) | I | 21 (7.0) | 17 (10.5) | 4 (3.5) | 0.089 |
| | II | 246 (82.0) | 139 (85.8) | 107(94.7) | |
| | III | 32 (10.7) | 5 (3.1) | 2 (1.8) | |
| | IV | 1 (0.4) | 1 (0.6) | 0 (0.0) | |
| Most recent CD4 count (cells/µl) (n = 272) | ≥500 | 102 (37.5) | 58 (35.6) | 44 (40.4) | 0.425 |
| | <500 | 170 (62.5) | 105 (64.4) | 65 (59.6) | |
| ART duration (months) | Mean (SD) | 18.15 ± 3.38 | 18.26 ± 3.24 | 17.97 ± 3.59 | 0.474 |
| Viral load suppression (n = 156) | Yes | 148 (94.9) | 79 (92.9) | 69 (97.2) | 0.293 |
| | No | 8 (5.1) | 6 (7.1) | 2 (2.8) | |

Note: ART: Antiretroviral therapy; SD: Standard deviation; WHO: World Health Organization.

## Sensitivity analysis

We examined the association between different cut-off thresholds of the baseline CD4 count and retention in care as well as the QoL. We found that retention rates did not significantly differ based on the baseline CD4 count thresholds, though a marginal association was observed at a higher cut-off. Similarly, mental and physical health QoL showed no significant differences across the baseline CD4 count categories (Table 4).

## Discussion

We determined if there is an association between baseline CD4 count and physical and mental QoL as well as retention in HIV care among PLWH aged ≥18 years. We found that ART initiation at a higher baseline CD4 count is associated with a lower mental health-related QoL but is not associated with physical health-related QoL and retention. The association between baseline CD4 count and QoL in PLWH has been a topic of huge interest over the years. Our study conducted during the UTT policy era showed that ART initiation at higher levels of baseline CD4 count is associated with a lower

**Table 2. Comparison of QoL with baseline CD4 count.**

| Quality of life (QoL) | Levels | CD4 < 500 cells/µl | CD4 ≥ 500 cells/ µl | P-value |
|---|---|---|---|---|
| Physical health-related QoL (n = 277) | Poor | 59 (36.0) | 27 (23.9) | 0.033 |
| | Good | 105 (64.0) | 86 (76.1) | |
| Mental health-related QoL (n = 81) | Poor | 2 (4.1) | 7 (21.9) | 0.025 |
| | Good | 47 (95.9) | 25 (78.1) | |
| Retained in HIV care (n = 300) | No | 9 (5.0) | 19 (15.7) | 0.002 |
| | Yes | 170 (95.0) | 102 (84.3) | |
| Follow-up time (months) | Median (IQR) | 17 (8-19) | 18 (9-20) | 0.290 |
| Mean physical health-related QoL | mean±SD | 66.1±1.74 | 66.7±2.59 | 0.015 |
| Mean mental health-related QoL | mean±SD | 133.6±1.64 | 133.8±1.46 | 0.586 |

**Note**: IQR: Interquartile range; SD denotes standard deviation; QoL: Quality of life.

**Table 3. Association between baseline CD4 count and QoL and retention in HIV care.**

| Primary outcome (+) | Levels | aOR (95% CI) |
|---|---|---|
| Physical health-related QoL | Poor | 1 |
| | Good | 0.54 (0.10-2.78) |
| Mental health-related QoL | Poor | 1 |
| | Good | 0.56* (0.32-0.97) |
| **Secondary outcome (++)** | | **aHR (95% CI)** |
| Retained in HIV care | No | 1 |
| | Yes | 0.30 (0.08-1.14) |

**Note:** (+) Measure of association was the odds ratio; (++) Measure of association was the hazard ratio; All odds ratios and hazard ratios have been adjusted for age, sex, most recent (current) CD4 count, antiretroviral (ART) duration, study site, and the World Health Organization (WHO) clinical stage. aOR: Adjusted odds ratio; aHR: Adjusted hazard ratio; CI: Confidence interval. All OR are exponentiated coefficients at a 5% significance level. Significance codes: * p < 0.05.

likelihood of good mental health-related QoL when compared with ART initiation at lower levels of baseline CD4 count. We found no difference in good physical health-related QoL between individuals who initiated ART at a higher baseline CD4 count compared to those who initiated ART at a lower baseline CD4 count. Previous studies, largely conducted before the UTT policy era show varying results regarding the association between CD4 count and QoL. Some of the past studies show a significant positive correlation between higher CD4 counts and better QoL [19], others suggest no direct relationship between changes in CD4 count and QoL scores in a Ugandan cohort of PLWH [20], and some report that CD4 count alone might be unreliable indicator of QoL [21].

Although the association is borderline statistically significant, the finding that PLWH with a higher baseline CD4 count are less likely to report good mental health-related QoL compared to those with a lower baseline CD4 count may be attributed to ART-related side effects. In general, the relationship between baseline CD4 count and mental health-related QoL is complex and is influenced by a combination of physical, psychological, and social factors.

ART exerts significant side effects on the brain, including drug toxicities and interactions hence affecting the mental health-related QoL of PLWH. A higher baseline CD4 count is often associated with a less advanced stage of HIV and better physical health. However, individuals with higher CD4 counts may still face significant psychological challenges. Despite their relatively better physical health, they may struggle with the emotional and psychological implications of living with HIV and life-long ART. These individuals may experience heightened anxiety about disease progression, uncertainty

**Table 4. Sensitivity analysis results between baseline CD4 count at different thresholds and retention and QoL.**

| Baseline CD4 cut-offs (cells/ul) | Level | Retention in care | | P-value |
|---|---|---|---|---|
| | | Yes (n = 272) | No (n = 28) | |
| At 200 baseline CD4 cut-off | CD4 < 200 | 36 (13.2) | 3 (10.7) | 0.706 |
| | CD4 ≥ 200 | 236 (86.8) | 25 (89.3) | |
| | | | | |
| At 350 baseline CD4 cut-off | CD4 < 350 | 99 (36.4) | 5 (17.9) | 0.050 |
| | CD5 ≥ 350 | 173 (63.6) | 23 (82.1) | |
| | | Mental Health QoL | | |
| | | Poor (n = 9) | Good (n = 72) | 0.440 |
| At 200 baseline CD4 cut-off | CD4 < 200 | 1 (11.1) | 16 (22.2) | |
| | CD4 ≥ 200 | 8 (88.9) | 56 (77.8) | |
| | | | | |
| At 350 baseline CD4 cut-off | CD4 < 350 | 3 (33.3) | 36 (50.0) | 0.345 |
| | CD5 ≥ 350 | 6 (66.7 | 36 (50.0) | |
| | | Physical Health QoL | | |
| | | Poor (n = 86) | Good (n = 191) | |
| At 200 baseline CD4 cut-off | CD4 < 200 | 12 (14.0) | 22 (11.5) | 0.568 |
| | CD4 ≥ 200 | 74 (86.10 | 169 (88.5) | |
| | | | | |
| At 350 baseline CD4 cut-off | CD4 < 350 | 31 (36.1) | 66 (34.5) | 0.810 |
| | CD5 ≥ 350 | 55 (63.9) | 125 (65.5) | |

regarding their long-term health, or the psychological burden of long-term ART. Moreover, they may be less likely to perceive themselves as "sick," potentially diminishing their sense of urgency about adhering to ART. This could create a paradox, where they feel less concerned about their health overall, even though HIV still requires ongoing management. Conversely, those with lower CD4 counts may have already encountered more severe health issues, which could foster better-developed coping strategies. Such individuals may have a clearer understanding of their health status and the critical need for adherence to ART, which could contribute to their enhanced mental well-being. Being more attuned to the immediate necessity of medication and healthcare, they may adopt a more proactive approach to health management, including seeking psychosocial support, thereby improving their overall mental health. Our findings thus highlight the significance of using appropriate screening tools and effective treatments for mental health issues among PLWH, the prioritization of the diagnosis and treatment of mental health disorders, and the integration of mental health screening and QoL measures in HIV programming.

We found good physical health-related QoL is similar between PLWH started on ART at a higher baseline CD4 count and those started on ART at a lower baseline CD4 count. This suggests that in resource-poor settings, health-related QoL can be greatly improved by effective ART as side effects are minimized so that PLWH achieve comparable physical health-related QoL [22]. A study in South Africa revealed that factors such as age, sex, ART-related side effects, and high pill burden reduce the QoL [23]. In addition, the impact of co-morbidities and pill burden on QoL becomes increasingly significant in PLWH as age advances. However, none of the measured demographic and behavioral factors was associated with QoL in our study.

The study showed no differences in retention in care between PLWH initiating ART at a higher baseline CD4 count and those initiating ART at a lower baseline CD4 count. This finding is consistent with our secondary hypothesis on retention in care.

However, a higher proportion of PLWH initiated on ART at a lower baseline CD4 (<500 cells/ul) are retained in HIV care compared to those initiated on ART at a higher baseline CD4 (≥500 cells/ul) but this difference disappeared after

controlling for confounders in the multivariable analysis. The finding agrees with previous study findings that show retention is higher among PLWH with lower baseline CD4 count compared to those with a higher baseline CD4 count. For instance, in a longitudinal retrospective cohort study involving PLWH on a second-line ART, a higher baseline CD4 cell count was associated with a lower retention in care [24]. In a South African study, results showed that loss to follow-up was high among PLWH started on ART at a higher baseline CD4 count (≥500 cells/ul) compared to those started on ART at a lower baseline CD4 (< 500 cells/ul).

Similarly, no significant association was observed between loss to follow-up and baseline CD4 count. Consistent with our findings, Chauke and colleagues [25] found that loss to follow-up was higher among PLWH started on ART under the UTT policy era with a higher baseline CD4 count (64.1%) compared to those started on ART at a lower baseline CD4 (35.9%). A randomized trial conducted in South Africa found that retention of PLWH initiated on ART at a higher baseline CD4 count under the UTT policy reached nearly 81% after 18 months of follow-up. However, factors such as younger age, recent HIV diagnosis, and male sex were associated with trial exit [26]. Also, PLWH initiating ART at a higher baseline CD4 count are at a higher risk for loss to follow-up as they consider themselves healthy [27].

Several factors contribute to retention in care among PLWH. A Zambian study reported individual-level barriers to retention in care, including ART-related side effects, mobility, weight gain, and interpersonal barriers such as stigma and nondisclosure of HIV status. Additionally, institutional-level barriers included inadequate clinic space, long waiting times, and longer travel distances to the HIV clinic [28]. In rural Uganda, undisclosed HIV serostatus has been shown to reduce adherence to clinic visits and presumably retention in the long term, including increasing clinic representation of PLWH [29]. The findings of the present study suggest a need to implement measures to optimize retention. For example, community and health facility-based differentiated ART delivery models are effective strategies for maximizing retention, as they ensure person-centered HIV care [30,31].

A Ugandan study showed that retaining largely asymptomatic PLHW in care is possible using strategies similar to those employed for persons who initiate ART at a higher baseline CD4 count such as serodiscordant couples [32]. Given the association between higher baseline CD4 counts and poor mental health-related QoL, HIV care programs should incorporate mental health services at ART initiation. The screening results would then guide referrals to mental health professionals for further assessment and intervention. Cognitive Behavioral Therapy (CBT)-based interventions may be incorporated into pre-ART counseling sessions to help PLWH cope with the psychological impact of HIV diagnosis and treatment initiation, especially for those initiating ART at high CD4 count. Additionally, peer support groups may help normalize the experience of initiating ART at higher CD4 counts, as this has been shown to reduce stigma and feelings of isolation. Health education on ART adherence and mental health should be included in HIV treatment literacy programs. Implementation research may be needed to evaluate the effectiveness of different mental health integration models in improving both mental health outcomes and retention in care. Adopting a comprehensive and patient-centered approach by the healthcare system may improve both mental health outcomes and long-term engagement in HIV care.

## Study strengths and limitations

Our study has several strengths and limitations. Regarding the strengths, few studies have compared QoL among PLWH initiating ART at higher versus lower CD4 counts in Uganda and across most HIV programs in sub-Saharan Africa, making our study findings novel. The use of baseline variables to demonstrate an association with the QoL ensured time sequence in our analysis, making the findings credible. The study was adequately powered to detect a statistically significant difference in the population. The data analyzed were collected using a validated and culturally appropriate tool with good psychometric properties (reliability and validity).

Limitations of the study include the lack of qualitative data to contextualize the findings as different factors might have shaped the QoL among PLWH in the UTT policy era. CD4 count was assessed at ART initiation. However, a comparison of CD4 count between the baseline and a follow-up period of up to 12 months might improve the interpretation of our

findings of the relationship between CD4 count and QoL. We studied retention over 20 months of follow-up and this may have not provided accurate results on its relationship with baseline CD4 count. A relatively longer follow-up time may be required in future studies.

Not all participants responded to the physical and mental health-related quality of life (QoL) domain questions, including a few independent variables resulting in missing data. This non-response may introduce selection bias, reduce statistical power, and limit the generalizability of the findings. Given its potential to compromise the reliability of the results, non-response must be carefully considered when interpreting the data. Moreover, we used a shorter time frame to measure retention compared to previous studies, which may have led to an overestimation of retention rates. However, the shorter time frame is more relevant to our study, as the maximum delay past a scheduled appointment was only 20 days. Potential unmeasured confounders, such as ART adherence, duration, regimen type, and socioeconomic factors, were not included in the study and should be considered when interpreting the results.

## Conclusions and recommendations

Our study showed that ART initiation at baseline CD4 count ≥500 cells/μl compared to baseline CD4 count <500 cells/μl is associated with a lower likelihood of good mental health-related QoL but similar good physical health-related QoL and retention in care. Our findings suggest a potential need for interventions to address mental health-related QOL, particularly among those initiating ART before symptoms emerge. Furthermore, person-centered interventions for ART delivery such as differentiated ART delivery models need to scale up to maintain retention in care.

## Supporting information

**S1 File. Dataset.**
(CSV)

## Acknowledgments

We thank TASO Mbarara for their support and PLWH who participated in the study. This dissertation arose from the Master of Public Health (MPH) dissertation for Ms Faith Tumuhairwe.

## Author contributions

**Conceptualization:** Faith Tumuhairwe, Anthanasio Bashaija, Francis Bajunirwe.

**Data curation:** Faith Tumuhairwe, Abel Munina, Anthanasio Bashaija, Francis Bajunirwe.

**Formal analysis:** Jonathan Izudi, Francis Bajunirwe.

**Funding acquisition:** Anthanasio Bashaija.

**Investigation:** Faith Tumuhairwe, Abel Munina, Anthanasio Bashaija, Francis Bajunirwe.

**Methodology:** Faith Tumuhairwe, Abel Munina, Anthanasio Bashaija, Francis Bajunirwe.

**Project administration:** Faith Tumuhairwe, Abel Munina, Anthanasio Bashaija.

**Resources:** Faith Tumuhairwe, Abel Munina, Anthanasio Bashaija.

**Software:** Jonathan Izudi, Abel Munina.

**Supervision:** Francis Bajunirwe.

**Validation:** Jonathan Izudi, Francis Bajunirwe.

**Visualization:** Jonathan Izudi, Francis Bajunirwe.

**Writing – original draft:** Faith Tumuhairwe, Jonathan Izudi, Abel Munina, Anthanasio Bashaija, Francis Bajunirwe.

**Writing – review & editing:** Faith Tumuhairwe, Jonathan Izudi, Abel Munina, Anthanasio Bashaija, Francis Bajunirwe.

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
