## [Decision Letter · Decision Letter 0]

10 Mar 2025

PONE-D-24-40617Quality of life and retention in care among people living with HIV initiated on ART in the era of “Universal Test and Treat” policy at a large HIV Clinic in South Western Uganda.PLOS ONE

Dear Dr. Izudi,

Thank you for submitting your manuscript to PLOS ONE. After careful consideration, we feel that it has merit but does not fully meet PLOS ONE’s publication criteria as it currently stands. Therefore, we invite you to submit a revised version of the manuscript that addresses the points raised during the review process.

We look forward to receiving your revised manuscript.

Kind regards,

Professor Kwasi Torpey, MD PhD MPH

Academic Editor

PLOS ONE

Journal Requirements:

2. We are unable to open your Supporting Information file S1 File.dta. Please kindly revise as necessary and re-upload.

Reviewers' comments:

Reviewer's Responses to Questions

**Comments to the Author**

1. Is the manuscript technically sound, and do the data support the conclusions?

Reviewer #1: Yes

Reviewer #2: Partly

2. Has the statistical analysis been performed appropriately and rigorously? 

Reviewer #1: Yes

Reviewer #2: No

3. Have the authors made all data underlying the findings in their manuscript fully available?

Reviewer #1: Yes

Reviewer #2: Yes

4. Is the manuscript presented in an intelligible fashion and written in standard English?

Reviewer #1: Yes

Reviewer #2: Yes

5. Review Comments to the Author

Reviewer #1: Strengths

● Setting of the study is relevant as it is in an area with high HIV burden in Africa so the results are generalizable

● They used a validated tool to measure QoL

● Good description of statistical analysis

● Sample size calculations well described

Weaknesses

● Wording in the abstract and part of the manuscript is contradictory- looking at the results on line 36 contradicts the findings on line 41

● The stratification of CD4 count into only <500 and ≥500 cells/μl oversimplifies the immunological response spectrum. It would have been more robust if there were finer CD4 categorization.

● With universal test and start the usefulness of the study is in question

● Only 81 (27%) of the participants responded to the mental health QoL domain with 277 responding to the physical health QoL however the authors did not explain the reason for the variance calling into question selection bias and the validity of the results

● The definition of retention (active in care ≤7 days after appointment vs. lost to follow-up >7 days after appointment) is quite strict compared to what we would have expected them to use e.g., WHO, definitions (often 30-90 days). The authors should justify this definition and discuss how it might affect comparability with other studies.

● The authors conclude that PLWH with higher baseline CD4 counts have lower mental health-related QoL, but the adjusted odds ratio of 0.56 (95% CI 0.32-0.97) has a confidence interval that nearly includes 1, suggesting a relatively weak association. This should be acknowledged in the discussion. This could be attributed to the sample size

● In lines 268-269, the authors report that the study findings concur with previous reports of , “ significant decrease in mental disorders among PLWH and but mild to moderate cognitive impairment remains high at all stages of HIV infection “ the results presented do not support this conclusion as the MH QoL does not specify what was assessed and the outcomes of the assessments.

● Authors should use more graphs and tables to describe/discuss the results

● While the study adjusted for several variables, other critical confounders such as ART adherence, duration on ART, ART regimen type, and socioeconomic factors were not included in the model.

● Language

○ Some sentences are excessively long, making comprehension difficult.

○ Terms such as "CD4 count at ART initiation" and "baseline CD4 count" are used interchangeably. They should standardize the language

○ The discussion section reiterates findings without substantial analysis

○ There are occasional grammatical errors and awkward phrasings that could be improved.. Generally more proof reading is needed

Minor concerns

● In Table 1, there are some discrepancies in the total numbers for variables like WHO clinical stage (n=275) and most recent CD4 count (n=272) compared to the total sample (n=300). These missing data should be explained.

● Looking at the table some numbers do not add up, WHO clinical stage 3 for example

● The authors report mean physical and mental health-related QoL scores in Table 2, but it's unclear how these scores should be interpreted (e.g., what is the possible range, what constitutes a clinically significant difference).

Suggestions

1. Provide a thorough explanation for the low response rate on the mental health-related QoL domain and discuss the implications for the validity of the findings.

2. Analyze and report the association between time since ART initiation and QoL outcomes.

3. Include a more detailed description of the MOS-HIV tool, including the range of possible scores and what constitutes a clinically meaningful difference.

4. Include data /scores of the different domains assessed using the QoL tool and assess explore associations between CD4 accounts and the different domains

5. Address potential selection bias, as those retained in care until the interview period might have different characteristics than those lost to follow-up.

6. Consider conducting sensitivity analyses with different definitions of retention to assess the robustness of the findings.

7. Discuss the implications of the findings more thoroughly, particularly how mental health interventions might be integrated into HIV care programs for those initiating ART at higher CD4 counts.

Reviewer #2: 1. The manuscript is generally well-organized, with a clear structure and logical progression. The introduction provides a solid background, the methods are detailed, and the discussion contextualizes the findings. However, there are some concerns about the robustness of the data analysis and the conclusions drawn:

a. The manuscript introduction provides a good background but lacks a clear research gap statement.

b. The historical progression of HIV treatment guidelines is informative but too detailed given HIV has been with us for decades and some of it could be summarized in a table.

c. The topic is relevant, though similar studies have been conducted in other settings. The manuscript would benefit from emphasizing how this study adds new insights beyond existing literature.

d. The justification for using the specific QoL measurement tool (MOS-HIV) should be strengthened given there are a number of tools, so important to justify why this specific tool was utilized.

e. The methods section is concise but could explicitly mention sample size justification. It is not clear how the 300 sample size was arrived at. Were all clients enrolled in the two years enrolled or only those with baseline CD4?

f. From the results and discussion, the primary finding, that higher CD4 count at ART initiation is associated with lower mental health-related QoL, is not fully explored. The authors speculate that this is due to ART-related side effects, but no direct evidence such as qualitative interviews or neuropsychiatric assessments is provided to support the claim.

g. Retention in care is analyzed, but the sample size is relatively small (n=300), compared to the clients in the clinic and retention is assessed over a relatively short period. A longer follow up period would give more reliable conclusions. Further, would be good to look at different time periods like 3, 6, 12 ,24 months retention.

h. There is no significant difference in physical health-related QoL or retention, but the authors still discuss the need for interventions to improve retention. While this is relevant, the justification based on the data provided is weak.

i. The categorization of CD4 count as <500 vs. ≥500 cells/µl used in the study is somewhat arbitrary. The manuscript references cut offs of 200, and 350 yet settles on 500 without a specific explanation why the 500 cut off. Especially given that quality of life at the different CD4 cut offs has been shown to be different including risk of those below 200 having high risk of advanced HIV disease compared to those above, sensitivity analyses using different cutoffs (e.g., <200 vs. ≥200, <350 vs. ≥350 ) could strengthen the findings.

j. The results section is repetitive—avoid re-explaining numbers already presented in tables.

k. The authors should discuss alternative explanations for their findings, particularly regarding mental health-related QoL.

l. There seems to be aspects of overinterpretation of results e.g the mental health QoL findings are interesting but should be cautious in attribution. The conclusion that higher CD4 count at ART initiation leads to lower mental health-related QoL is not fully supported by the data. Other attributes such as socioeconomic factors, mental health comorbidities, or ART side effects should be considered.

m. The conclusions in the manuscript align with the results, be more direct and linked to the study’s aims and findings.

n. There’s need to have the policy implications brought out clearly and recommendations for healthcare interventions based on findings. The authors discuss mental health interventions for early ART initiators, but specific recommendations are missing.

o. No mention of study limitations yet important to understand if any potential biases.

2. The authors have done a good analysis and used appropriate statistical models (logistic regression for QoL outcomes and Cox proportional hazards for retention). However, there are some concerns:

a. The assumptions for the Cox model (proportional hazards) are not explicitly tested. The authors should test proportional hazards assumptions for the Cox model.

b. The logistic regression model does not report any goodness-of-fit tests. This is necessary to confirm that the model is valid.

c. The statistical analyses are generally appropriate, but consider analyzing retention further with additional sensitivity checks e.g., competing risks analysis for mortality among those with very low CD4 counts vs. loss to follow-up especially those with high CD4s with the perception of “being well”.

d. The categorization of CD4 count into two groups only groups loses potentially valuable data. Will be better if considered e.g., quartiles or a continuous variable to give reveal additional trends.

e. The study uses a cross-sectional QoL assessment, but the authors attempt to infer causality (suggesting that higher CD4 at initiation leads to lower mental QoL). A longitudinal design would be more appropriate for such conclusions.

f. Tables should be simplified with clear footnotes explaining abbreviations.

g. If possible, analyze QoL differences by gender, age group, or socioeconomic factors to add depth to findings.

3. The manuscript states that all relevant data are fully available. A data availability statement is included, and the data seem to be within the manuscript and supporting files. However, The authors should consider depositing their dataset in a public repository with a DOI for transparency.

4. The manuscript is generally well-written in standard English, but there are some grammatical errors, phrasing that can be improved and redundancies that need to be addressed.

a. Typographical errors noted in a number of areas. E.g lines 51-52: “Globally, there were 39 million people were living with human immunodeficiency virus (PLWH) by the end of 2022 [1].” Has a “were.” Highlighted that is unnecessary

Lines 53-54: “HIV is a treatable and preventable condition with life-long antiretroviral therapy (ART) and its scale-up has led to HIV becoming a manageable chronic health condition, enabling PLWH to live longer and healthier lives, almost comparable to people without HIV.”

The part of sentence “its scale-up has led to HIV becoming…” is off as it makes the sentence read like HIV scale up yet I believe reference is to ART thus could read better as “HIV is a treatable and preventable condition with lifelong antiretroviral therapy (ART). Its widespread use has transformed HIV into a manageable chronic condition, allowing PLWH to live longer and healthier lives, similar to those without HIV.” Line 282-283:

“A study in South Africa revealed that factors such as age, sex, ART, and pill burden reduce the QoL [24].” To note is, “ART” is not a factor that reduces QoL; it should specify “ART-related side effects.”

b. Some sentences are overly complex, making the manuscript harder to follow. For example, lines 69-72 “Early ART initiation (regardless of immune and clinical status) has several clinical and public health benefits such as preventing the progression of HIV to AIDS and preventing its transmission to other individuals as PLWH with undetectable viral load do not transmit HIV dubbed as Undetectable = Untransmittable or U=U [3, 7, 8].” Is too long with lots of information in one sentence, the phrase dubbed doesn’t sound good for a manuscript, and the “Undetectable = Untransmittable” (U=U) concept needs clearer introduction. Consider breaking it for example to read as “Early ART initiation, regardless of immune and clinical status, has multiple benefits. It helps prevent the progression of HIV to AIDS and reduces transmission. Research shows that PLWH with an undetectable viral load cannot transmit HIV, a concept known as ‘Undetectable = Untransmittable’ (U=U) [3,7,8].”

c. Generally, edit as you simplify the long/complex sentences to improve readability.

d. The results section repeats information that is already provided in tables, leading to redundancy. Consider having information in tables not repeated again in the narrative unless giving further details on the results. Instead of repeating full results, focus on key statistical takeaways.

e. Some references are outdated and should be replaced with recent studies (e.g., WHO 2024 guidelines).

6. PLOS authors have the option to publish the peer review history of their article (what does this mean? ). If published, this will include your full peer review and any attached files.

**Do you want your identity to be public for this peer review?** For information about this choice, including consent withdrawal, please see our Privacy Policy .

Reviewer #1: No

Reviewer #2: No

---

## [Author Response · Author response to Decision Letter 1]

15 Apr 2025

Point-by-point responses to reviewer’s comments

We sincerely thank the Associated Editor, Prof. Kwasi Torpey, for overseeing the review process and the reviewers for their insightful and constructive comments, which have significantly improved the quality of our manuscript. We have carefully addressed all reviewers’ comments and provided detailed responses, indicating exactly how each point has been incorporated into the revised manuscript. We look forward to your favorable response.

Reviewer #1:

Strengths

● Setting of the study is relevant as it is in an area with high HIV burden in Africa so the results are generalizable

● They used a validated tool to measure QoL

● Good description of statistical analysis

● Sample size calculations well described

Response: Thank you for the encouraging comments.

Weaknesses

● Wording in the abstract and part of the manuscript is contradictory- looking at the results on line 36 contradicts the findings on line 41

Response: We agree with this observation. The results and conclusions have been harmonized. The conclusion now reads: “In this cohort of PLWH on ART, those who initiated the treatment at a higher baseline CD4 were less likely to have good mental health-related QoL functioning compared to those initiating at lower CD4 counts. However, the two groups were comparable in physical health-related QoL and retention in care.”

● The stratification of CD4 count into only <500 and ≥500 cells/μl oversimplifies the immunological response spectrum. It would have been more robust if there were finer CD4 categorization.

Response: A more granular categorization of CD4 counts could provide deeper insights into associations. However, such disaggregation may lead to information loss and reduced statistical power. Clinically, CD4 count is commonly categorized as <500 and ≥500 cells/µL to reflect immunological status, as lower counts (typically below 500 cells/µL) are associated with WHO clinical stage 3 opportunistic infections. We have therefore maintained the categorization for these reasons.

● With universal test and start the usefulness of the study is in question

Response: Universal test and treat (UTT) enables ART initiation immediately after HIV diagnosis, regardless of CD4 count. We thank the reviewer for raising this point, and indeed we are aware it is now the standard of care, and this is the basis of our research question. There are concerns about impact of universal test and test on quality of life (QoL) and retention, as long-term ART exposure may increase the risk of adverse effects, drug toxicities, and interactions, especially among persons initiating ART at higher CD4 counts, i.e. without symptoms of HIV disease. To our knowledge, previous studies have primarily examined QoL among individuals with low CD4 counts. Providing evidence on QoL in those with high CD4 counts has important clinical and public health implications, as discussed in the introduction (page 3, paragraphs 2–3).

● Only 81 (27%) of the participants responded to the mental health QoL domain with 277 responding to the physical health QoL however the authors did not explain the reason for the variance calling into question selection bias and the validity of the results

Response: We are grateful for this comment and this non-response has been reported in the results section previously. In the revised manuscript, we have acknowledged the non-response rate as one of the study limitations. The new text reads as follows:

“Not all participants responded to the physical and mental health-related quality of life (QoL) domain questions, including a few independent variables resulting in missing data. This non-response may introduce selection bias, reduce statistical power, and limit the generalizability of the findings. Given its potential to compromise the reliability of the results, non-response must be carefully considered when interpreting the data.

● The definition of retention (active in care ≤7 days after appointment vs. lost to follow-up >7 days after appointment) is quite strict compared to what we would have expected them to use e.g., WHO, definitions (often 30-90 days). The authors should justify this definition and discuss how it might affect comparability with other studies.

Response: We agree that a strict definition was used in the present study compared to previous studies and this may have led to overestimation. The timeline is appropriate in our study as the maximum number of days elapsed following the scheduled appointment was just 20 days. This suggests that the 30-day rule was not applicable in the study. We have acknowledged this in the limitations section. The text reads as follows:

“Moreover, we used a shorter time frame to measure retention compared to previous studies, which may have led to an overestimation of retention rates. However, the shorter time frame is more relevant to our study, as the maximum delay past a scheduled appointment was only 20 days”.

● The authors conclude that PLWH with higher baseline CD4 counts have lower mental health-related QoL, but the adjusted odds ratio of 0.56 (95% CI 0.32-0.97) has a confidence interval that nearly includes 1, suggesting a relatively weak association. This should be acknowledged in the discussion. This could be attributed to the sample size

Response: We agree. A new text regarding the attenuated association has been added and it reads:

“Although the association is borderline statistically significant, the finding that PLWH with a higher baseline CD4 count are less likely to report good mental health-related QoL compared to those with a lower baseline CD4 count may be attributed to ART-related side effects.”

● In lines 268-269, the authors report that the study findings concur with previous reports of , “ significant decrease in mental disorders among PLWH and but mild to moderate cognitive impairment remains high at all stages of HIV infection “ the results presented do not support this conclusion as the MH QoL does not specify what was assessed and the outcomes of the assessments.

Response: This sentence has been deleted.

● Authors should use more graphs and tables to describe/discuss the results

Response: Thank you for your suggestion to use more graphs and tables to describe and discuss the results. We have presented our results in four tables and believe this represents the average number of tables in a manuscript. However, we are open to revisiting this if specific results could be more effectively conveyed through graphs.

● While the study adjusted for several variables, other critical confounders such as ART adherence, duration on ART, ART regimen type, and socioeconomic factors were not included in the model.

Response: We agree that there are unmeasured confounders and this has been included in the study limitations section. The text reads as follows:

“Potential unmeasured confounders such as ART adherence, duration, regimen type, and socioeconomic factors, were not included in the study and should be considered when interpreting the results.”

● Language

○ Some sentences are excessively long, making comprehension difficult.

Response: We have re-checked and shortened longer sentences.

○ Terms such as "CD4 count at ART initiation" and "baseline CD4 count" are used interchangeably. They should standardize the language

Response: Thank you for the comment. In the revised manuscript, we used “Baseline CD4 count”.

○ The discussion section reiterates findings without substantial analysis

Response: We have revised most texts in the discussion to ensure substantial analysis.

○ There are occasional grammatical errors and awkward phrasings that could be improved.. Generally more proofreading is needed

Response: We have proofread the manuscript and corrected all grammatical errors and awkward phrasings.

Minor concerns

● In Table 1, there are some discrepancies in the total numbers for variables like WHO clinical stage (n=275) and most recent CD4 count (n=272) compared to the total sample (n=300). These missing data should be explained.

Response: We acknowledge that some data is missing, and this issue has been addressed in the limitations section. For more details, please refer to our response to comment #4.

● Looking at the table some numbers do not add up, WHO clinical stage 3 for example

Response: This has been addressed in the responses above.

● The authors report mean physical and mental health-related QoL scores in Table 2, but it's unclear how these scores should be interpreted (e.g., what is the possible range, what constitutes a clinically significant difference).

Response: The physical and mental health-related QoL scores based on the Medical Outcomes Study-HIV Health Survey (MOS-HIV) tool range from 0 to 100. We added the statement 'The scores range from 0 to 100' under the 'Variables and Measurement' sub-section in the Methods and Materials section. Since the scores are numerical, we assessed whether statistically significant differences in mean scores existed based on baseline CD4 count categorization. To our knowledge, no established threshold exists for defining a clinically significant difference in mean scores.

Suggestions

1. Provide a thorough explanation for the low response rate on the mental health-related QoL domain and discuss the implications for the validity of the findings.

Response: We have addressed the non-response rate and included its implication on the overall results.

2. Analyze and report the association between time since ART initiation and QoL outcomes.

Response: We are grateful for this comment. The time since ART initiation was included as a covariate in the analysis of the relationship between baseline CD4 count and QoL. Please see the footnote in Table 3.

3. Include a more detailed description of the MOS-HIV tool, including the range of possible scores and what constitutes a clinically meaningful difference.

Response: There is no universally agreed-upon threshold for a clinically meaningful difference in scores from the Medical Outcomes Study-HIV Health Survey (MOS-HIV). However, based on similar health-related quality of life (HRQoL) instruments, researchers often consider a difference of 5–10 points to be clinically significant. In response to the comment, we have added the text below in the methods section:

"While no universally accepted threshold exists for a clinically meaningful difference in MOS-HIV scores, prior research on similar Health-Related Quality of Life (HRQoL) tools suggests that differences in the range of 5–10 points may be considered clinically significant."

4. Include data /scores of the different domains assessed using the QoL tool and assess explore associations between CD4 accounts and the different domains

Response: We chose to present the summary scores as they are simpler, rather than the individual specific domains as they are over 10 and we did not have specific hypothesis on differences among the domains.

5. Address potential selection bias, as those retained in care until the interview period might have different characteristics than those lost to follow-up.

Response: We recognize some PLWH are lost to follow-up during care, before they were interviewed. However, given that retention was similar by level of CD4 count at treatment initiation, suggests that selection bias may not play a significant role.

6. Consider conducting sensitivity analyses with different definitions of retention to assess the robustness of the findings.

Response: The maximum delay past a scheduled appointment was only 20 days in our study so we did not perform a sensitivity analysis based on a much shorter day.

7. Discuss the implications of the findings more thoroughly, particularly how mental health interventions might be integrated into HIV care programs for those initiating ART at higher CD4 counts.

Response: We have provided additional text around the implications of findings and it reads:

“Given the association between higher baseline CD4 counts and poorer mental health-related QoL, HIV care programs should incorporate mental health services at the point of ART initiation. The screening results would then guide referrals to mental health professionals for further assessment and intervention. Cognitive Behavioral Therapy (CBT)-based interventions may be incorporated into pre-ART counseling sessions to help PLWH cope with the psychological impact of HIV diagnosis and treatment initiation, especially for those initiating ART at high CD4 count. Additionally, peer support groups may help normalize the experience of initiating ART at higher CD4 counts, as this has been shown to reduce stigma and feelings of isolation. Health education on ART adherence and mental health should be included in HIV treatment literacy programs. Implementation research may be needed to evaluate the effectiveness of different mental health integration models in improving both mental health outcomes and retention in care. Adopting a comprehensive and patient-centered approach by the healthcare system may improve both mental health outcomes and long-term engagement in HIV care.”

Reviewer #2:

1. The manuscript is generally well-organized, with a clear structure and logical progression. The introduction provides a solid background, the methods are detailed, and the discussion contextualizes the findings. However, there are some concerns about the robustness of the data analysis and the conclusions drawn:

a. The manuscript introduction provides a good background but lacks a clear research gap statement.

Response: Thank you for this comment. We have re-checked the introduction section and have re-affirmed the problem statement/research gap as indicated in the introduction section (paragraph 4, page 4). The gap reads:

“Therefore, it is not completely clear whether ART initiation at a higher baseline CD4 count might lead to negative effects on QoL or retention compared to ART initiation at a lower baseline CD4 count in PLWH in resource-limited settings such as Uganda”.

b. The historical progression of HIV treatment guidelines is informative but too detailed given HIV has been with us for decades and some of it could be summarized in a table.

Response: We have summarized the information about the HIV treatment guidelines. We aim to provide clear information about changes in treatment guidelines and how this has impacted care delivery, which is important in understanding the need for the current research.

c. The topic is relevant, though similar studies have been conducted in other settings. The manuscript would benefit from emphasizing how this study adds new insights beyond existing literature.

Response: Previous studies have examined QoL at lower CD4 counts. Therefore, the relationship between ART initiation at a higher baseline CD4 count and QoL or retention in resource-limited settings such as Uganda remains completely unclear. Our study filled this gap and this evidence has important clinical and public health implications as discussed in the introduction (page 3, paragraphs 2–3).

d. The justification for using the specific QoL measurement tool (MOS-HIV) should be strengthened given there are a number of tools, so important to justify why this specific tool was utilized.

Response: We have included a text summarizing the rationale for using the MOS-HIV tool for QoL measurement on page 5 (variable and measurement sub-section).

e. The methods section is concise but could explicitly mention sample size justification. It is not clear how the 300 sample size was arrived at. Were all clients enrolled in the two years enrolled or only those with baseline CD4?

Response: The sample size determination was hypothesis-driven, as described on page 6 in the sample size calculation section. Given that this was an analytical study, we powered the study at 80% to detect the anticipated mean difference between the comparison groups. The two-year period provided the sampling frame; however, not all individuals within this timeframe were included, as we used consecutive sampling. For each sampled individual, we retrieved baseline CD4 count data. These issues have been explained in the respective sections of the methods.

f. From the results and discussion, the primary finding, that higher CD4 count at ART initi

---

## [Editor Report · Decision Letter 1]

20 Apr 2025

Quality of life and retention in care among people living with HIV initiated on ART in the era of “Universal Test and Treat” policy at a large HIV Clinic in South Western Uganda.

PONE-D-24-40617R1

Dear Dr. Izudi

We’re pleased to inform you that your manuscript has been judged scientifically suitable for publication and will be formally accepted for publication once it meets all outstanding technical requirements.

Kind regards,

Professor Kwasi Torpey, MD PhD MPH

Academic Editor

PLOS ONE

Additional Editor Comments (optional):

The comments to the authors have been adequately addressed. Where there are differences in opinion with reviewer comments, the explanation provided is satisfactory and reasonable
---

## [Editor Report · Acceptance letter]

PONE-D-24-40617R1

PLOS ONE

Dear Dr. Izudi,

I'm pleased to inform you that your manuscript has been deemed suitable for publication in PLOS ONE. Congratulations! Your manuscript is now being handed over to our production team.

Kind regards,

on behalf of

Professor Kwasi Torpey

Academic Editor

PLOS ONE